# Age-related slowing down in the motor initiation in elderly adults

**Nikita S. Frolov**[1*], **Elena N. Pitsik**[1], **Vladimir A. Maksimenko**[1], **Vadim V. Grubov**[1,2], **Anton R. Kiselev**[2], **Zhen Wang**[3], **Alexander E. Hramov**[1,2]

**1** Neuroscience and Cognitive Technology Laboratory, Innopolis University, Innopolis, The Republic of Tatarstan, Russia, **2** Saratov State Medical University, Saratov, Russia, **3** Northwestern Polytechnical University, Xi'an, Shaanxi, China

☉ These authors contributed equally to this work.
* n.frolov@innopolis.ru

**Data Availability Statement:** The data underlying the results presented in the study is available at https://doi.org/10.6084/m9.figshare.12301181.

**Funding:** NF received funding for this work from the Council on grants of the President of the

## Abstract

Age-related changes in the human brain functioning crucially affect the motor system, causing increased reaction time, low ability to control and execute movements, difficulties in learning new motor skills. The lifestyle and lowered daily activity of elderly adults, along with the deficit of motor and cognitive brain functions, might lead to the developed ambidexterity, i.e., the loss of dominant limb advances. Despite the broad knowledge about the changes in cortical activity directly related to the motor execution, less is known about age-related differences in the motor initiation phase. We hypothesize that the latter strongly influences the behavioral characteristics, such as reaction time, the accuracy of motor performance, etc. Here, we compare the neuronal processes underlying the motor initiation phase preceding fine motor task execution between elderly and young subjects. Based on the results of the whole-scalp sensor-level electroencephalography (EEG) analysis, we demonstrate that the age-related slowing down in the motor initiation before the dominant hand movements is accompanied by the increased theta activation within sensorimotor area and reconfiguration of the theta-band functional connectivity in elderly adults.

## Introduction

Healthy aging affects neural processes by changing the neurochemical and structural properties of the brain [1]. It determines the cognitive and motor performance decline during a daily activity of elderly adults and negatively influences the quality of their life. The markers of age-related neural impairments are observed at the behavioral level as slowing of the reaction time (RT), reduced motor control and coordination, etc. [2, 3].

Upper limbs represent the most active part of the human motor system; thus, the degradation of its functioning with age is the most prominent [4]. Plenty of studies report difficulties in accomplishing complex motor tasks related to the deficit of hand movement coordination, ability to control force, execute sequential actions, learn new motor skills, etc. [2, 5, 6]. The motor performance decline while executing fine motor tasks is also well-documented [7, 8]. Several studies report that the level of beta-band (15-30 Hz) oscillations is a relevant marker of

Russian Federation (Grant No. MK-2080.2020.2). AH received funding for this work from the Russian Foundation for Basic Research (Grant No. 19-52-55001) and the Council on grants of the President of the Russian Federation (Grant No. NSh-2594.2020.2). The funders had no role in study design, data collection and analysis, decision to publish, or preparation of the manuscript.

**Competing interests:** The authors have declared that no competing interests exist.

a decreased motor performance in healthy aging and disease [9–15]. Particularly, an increased movement-related beta desynchronization (MBRT) was linked with a greater GABAergic inhibitory activity in the primary motor cortex and suggested to influence the motor plasticity of elderly subjects [9–11]. A tentative relation between peri-movement beta-band desynchronization and motor performance was shown in [12, 13]. Also, a decreased post-movement beta rebound (PMBR) within the medial prefrontal cortex of elderly adults indicated an impaired cognitive control of stimulus-induced motor tasks [14].

Besides, a broad literature link the age-related motor performance decline with an over-activation of the motor and prefrontal area of the human brain, which control the motor execution process [16–19]. Specifically, the recruitment of additional ipsilateral motor regions in elderly adults is supposed to provide a compensatory mechanism that supports overcoming the age-related structural changes in the human brain [20, 21]. On the one hand, it helps to maintain the performance of executed motor actions. On the other hand, this mechanism demands more neuronal resources and, therefore, slows the motor response. Also, several studies relate the over-activation of cortical areas to the 'use-dependent plasticity' [22], which is supposed to underlie dedifferentiation of brain functions in advanced age. In the context of the motor system, it is manifested as a developed ambidexterity, i.e., a loss of the dominant limb advances [8, 23].

While the age-related differences cortical activation directly related to motor execution and control is extensively studied, less is known about the effect of healthy aging on the motor planning phase and its influence on RT. Exploring these mechanisms is crucial to deeper understand motor control in humans. Motor planning is also subjected to the age-related changes due to the following: (i) motor initiation process involves many higher cognitive functions such as sensory processing, working memory, motor embodiment, and sensorimotor integration [24–27], which are known to decline strongly with age; (ii) the theta activity underlying the majority of these processes exhibits significant age-related changes—abnormally increased theta activity in elderly people indicates subjective cognitive dysfunction and suspected dementia [28, 29].

Based on the above, we hypothesize that the age-related changes in the motor planning mechanism also affect the slowing of the motor initiation phase in elderly adults. To address the issue, we considered the differences in cortical activity during the controlled execution of fine motor tasks between elderly adults and young adults using electroencephalography (EEG). Consistent with the dedifferentiation theory [8, 23], we found that the motor cortex of younger adults activated much faster during the dominant hand task, while in elderly adults, the time required for motor activation was equal for both hands and approached the level of the non-dominant hand of younger adults. Further, as expected, we found the significant differences in cortical activation during the time interval preceding the motor action. In elderly adults, as well as in young adults performing the non-dominant hand task, we observed the increased theta-band power in the frontal, central, and central-parietal EEG sensor rows, whereas theta-activation was insignificant in young adults during the dominant hand task. Finally, based on the results of between-subject functional connectivity analysis, we revealed that motor planning involves different types of cortical interactions in young adults and elderly adults.

## Materials and methods

### Participants

Two groups of healthy volunteers, including 10 elderly adult subjects (EA group; age: 65±5.69 (MEAN±SD); range: 55-72; 4 males, 6 females) and 10 young adult subjects (YA group; age:

26.1±5.15 (MEAN±SD); range: 19-33; 7 males, 3 females), participated in this study. All subjects were right-handed and had no history of brain tumors, trauma or stroke-related medical conditions. The experimental protocol was approved by the local research Ethics Committee of Innopolis University. The experimental study was performed in accordance with the Declaration of Helsinki. All participants were pre-informed about the goals and design of the experiment and signed a written informed consent.

## Task

All participants were instructed to sit on the chair with their hands lying comfortably on the table desk in front of them, palms up. The timeline of the experimental session is presented in Fig 1A. First, we recorded Eyes Open Resting state (5 minutes). Further experiment included sequential repetitions of the fine motor task (squeezing one of the hands into a fist after the audio signal and holding it until the second signal) using either left or right hand (30 repetitions per hand, 60 in total). The duration of the signal determined the type of movement: short beep (0.3 s) was given to perform a non-dominant hand (left hand, LH) movement and long beep (0.75 s) was given to perform a dominant hand (right hand, RH) movement. Thus, we conducted a mixed-design experimental study with the Movement Type (LH and RH conditions) as within-subject factor and the Age (EA and YA groups) as between-subject factor.

The timeline of a single motor task is presented in Fig 1B. The time interval between the signals during the task and the pause between the repetitions were chosen randomly in the range 4–5 s and 6–8 s, respectively. The types of movements (LH or RH) were mixed in the course of the session and given randomly to exclude possible training or motor-preparation effects caused by the sequential execution of the same tasks. The overall experimental session lasted approximately 16 minutes, including the background cortical activity recording and series of movement executions.

## EEG data acquisition and preprocessing

We acquired EEG signals using the monopolar registration method (a 10—10 system proposed by the American Electroencephalographic Society [30]). According to this, we recorded EEG signals with 31 sensors (O2, O1, P4, P3, C4, C3, F4, F3, Fp2, Fp1, P8, P7, T8, T7, F8, F7, Oz, Pz, Cz, Fz, Fpz, FT7, FC3, FCz, FC4, FT8, TP7, CP3, CPz, CP4, TP8) and two reference electrodes A1 and A2 on the earlobes and a ground electrode N just above the forehead. We used the cup adhesive Ag/AgCl electrodes placed on the "Tien–20" paste (Weaver and Company, Colorado, USA). Immediately before the experiments started, we performed all necessary procedures to increase skin conductivity and reduce its resistance using the abrasive "NuPrep" gel (Weaver and Company, Colorado, USA). We controlled the variation of impedance within a range of 2–5 kΩ during the experiment. The electroencephalograph "Encephalan-EEG-19/26" (Medicom MTD company, Taganrog, Russian Federation) with multiple EEG and two EMG channels performed amplification and analog-to-digital conversion of the recorded signals. The EMG signals were acquired to verify the correctness of the epochs segmentation. This device possessed the registration certificate of the Federal Service for Supervision in Health Care No. FCP 2007/00124 of 07.11.2014 and the European Certificate CE 538571 of the British Standards Institute (BSI).

The raw EEG and EMG signals were sampled at 250 Hz and filtered by a 50–Hz notch filter by embedded hardware-software data acquisition complex. Additionally, raw EEG signals were filtered by the 5th-order Butterworth filter with cut-off points at 1 Hz and 100 Hz. Eyes blinking and heartbeat artifact removal was performed by the Independent Component Analysis (ICA) [31]. The recorded EEG and EMG signals presented in proper physical units

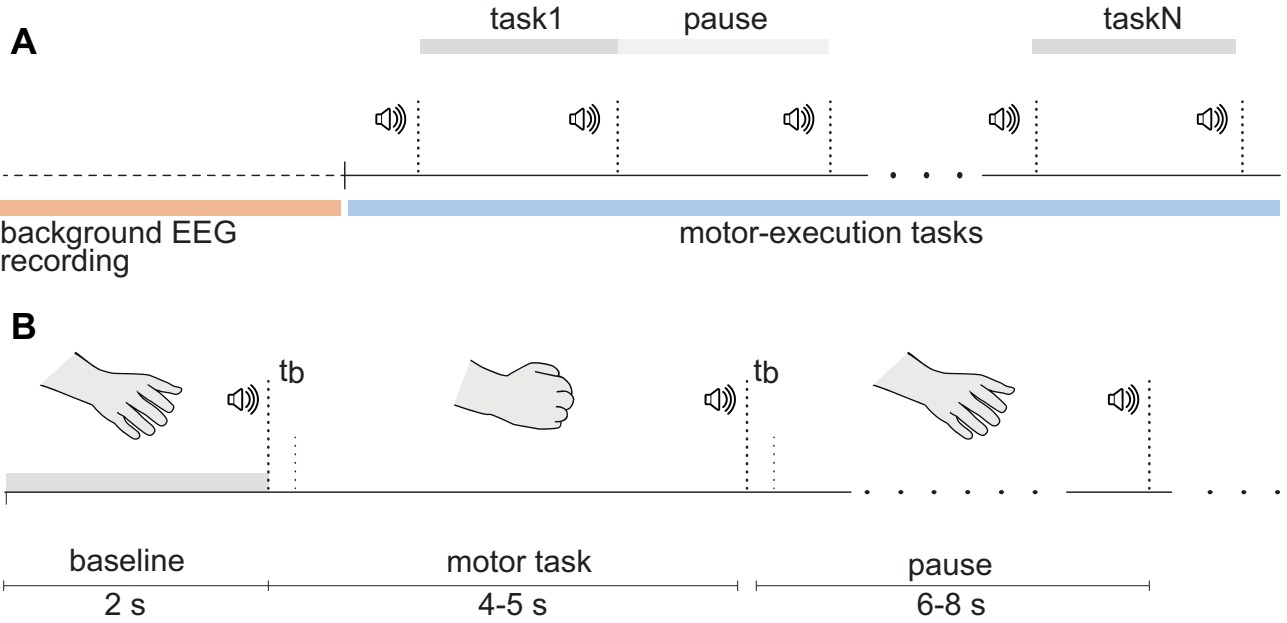

**Fig 1. Experimental paradigm.** Timelines of the experimental session (**A**) and a single motor task (**B**). Here, $t_b$ is the duration of the beep, which is 0.3 s for the LH movement command, and 0.75 s for the RH movement command.

(millivolts) were segmented into four sets of epochs according to the (group [YA, EA], condition [LH, RH]) combinations: YA LH, YA RH, EA LH, and EA RH. Each epoch was 10 s long, including 2s baseline activity and 8s motor-related activity. Data was then inspected manually and corrected for remaining artifacts. Epochs which we failed to correct manually mostly due to the strong muscle artifacts were rejected. Finally, each set contained 15 corrected epochs, which was equal to the minimal number of the artifact-free epochs over all participants.

All preprocessing steps including filtering, artifact removal and epoching were performed using MNE package (ver. 0.20.0) for Python 3.7 [32]. The analyzed EEG data is available online [33].

### Time-frequency analysis in sensor space

For each (group, condition)–set of epochs, we estimated spectral power in theta (4-8 Hz), alpha/mu (8-14 Hz) and beta (15-30 Hz) frequency bands using time-frequency analysis implemented in MNE. Particularly, time-frequency representation of the EEG epochs was obtained via Morlet complex-valued wavelet in the range 4-30 Hz and contrasted with 2s baseline period using 'percent' mode, i.e., subtracting the mean of baseline values followed by dividing by the mean of baseline values. The number of cycles in the wavelet transform was set for each frequency $f$ as $f/2$. Then, obtained time-frequency representations were averaged over epochs for each subject.

**Estimation of the motor brain response time.** A priory knowledge about the cortical activation during actual movements execution implies that motor brain response is determined as a pronounced event-related desynchronization (ERD) of mu and beta oscillations in the contralateral area of the motor cortex [34–39]. Here, we used mu- and beta-band event-related spectral power ($ERSP_{\mu,\beta}$) at C4 sensor in LH condition and C3 sensor in RH condition to estimate motor brain response time in corresponding frequency bands ($MBRT_{\mu,\beta}$) for each subject of both groups. We manually inspected each $ERSP_{\mu,\beta}$ time-series and defined $MBRT_{\mu,\beta}$

as the first minimum of the spectral power below the 2.5th baseline level (an exemplary illustration of the $\text{MBRT}_\mu$ estimation is presented in Fig 2A). Thus, we collected eight sets of MBRT corresponding to each (group, condition, frequency band)-set. Statistical comparison of the MBRT was performed using a two-way mixed-design ANOVA test implemented in JASP open-source statistical software [40].

**Within-subject time-frequency analyses.**   We performed within-subject spatio-temporal clustering analyses to reveal arrays of sensors associated with the motor-related brain activity separately in each frequency band of interest for each age group and experimental condition. Pairwise comparison of (time,sensor)-pairs was performed via one-tailed one-sampled $t$-test ($dF = 9$, $p_{pairwise} = 0.005$, $t_{critical} = \pm 3.2498$) and spatio-temporal clustering was assessed using non-parametric permutation test with $r = 2000$ random permutations ($p_{cluster} = 0.05$) following Maris and Oostenveld [41].

**Between-subject time-frequency analyses.**   During between-subject analyses, we compared brain activity of the age groups in the same experimental conditions. Again, we considered baseline-corrected topographic maps averaged in the frequency bands of interest. Effect of interest was evaluated at each (time,sensor)–pair using one-tailed unpaired $F$-test for independent samples ($dF1 = 1$, $dF2 = 18$, $p = 0.025$, $F_{critical} = 10.218$) and spatio-temporal clustering was assessed using non-parametric permutation test with $r = 2000$ random permutations ($p_{cluster} = 0.05$) [41].

**Mixed-design analyses.**   Based on the results of within- and between-subject spatio-temporal clustering analyses, we localized the effect of significant spectral power change in the spatio-temporal domain. Further, for each (group, condition)-set we averaged spectral power over the corresponding spatio-temporal clusters and compared it using mixed-design ANOVA.

## Functional connectivity analysis

Functional connectivity measures the similarity of activation in the different brain regions based on the recorded signals of brain activity. According to the review papers [42, 43], there exists a variety of functional connectivity metrics that evaluate this similarity in the different aspects. Moreover, functional connectivity analysis based on EEG or MEG recordings suffers from such problems as volume conduction/field spread effect, signal-to-noise ratio, common input, etc. due to the nature of these neuroimaging techniques [44]. Thus, the choice of the particular functional connectivity measure requires both a prior knowledge about the analyzed neuronal processes and an understanding of possible problems that may potentially interfere with the adequate interpretation of functional connectivity results.

**Functional connectivity measure.**   In accordance with the prior knowledge that motor-related activity is associated with certain frequency bands, first of all we expect the similarity of oscillatory behavior in remote brain regions in terms of phase-locking. Among the variety of FC measures based on the phase-synchronization, phase lag index (PLI) seems to be an appropriate metric [45]. PLI is robust to the common source problem as it ignores simultaneous phase similarity, less sensitive to the intrinsic EEG noise and allows reasonable interpretation of the obtained results. PLI is traditionally defined as:

$$\text{PLI}_{i,j} = |\langle \text{sign}(\phi_i(t_k) - \phi_j(t_k)) \rangle|, \tag{1}$$

where $\phi_{i,j}(t)$ are phases of signals at $i^{th}$ and $j^{th}$ EEG sensors introduced via Hilbert transform and operator $\langle \bullet \rangle$ averaging over time points $k$. It clearly follows from Eq (2), that PLI lies between 0 and 1, where PLI = 1 corresponds to perfect phase-locking and PLI = 0 implies a complete lack of synchrony.

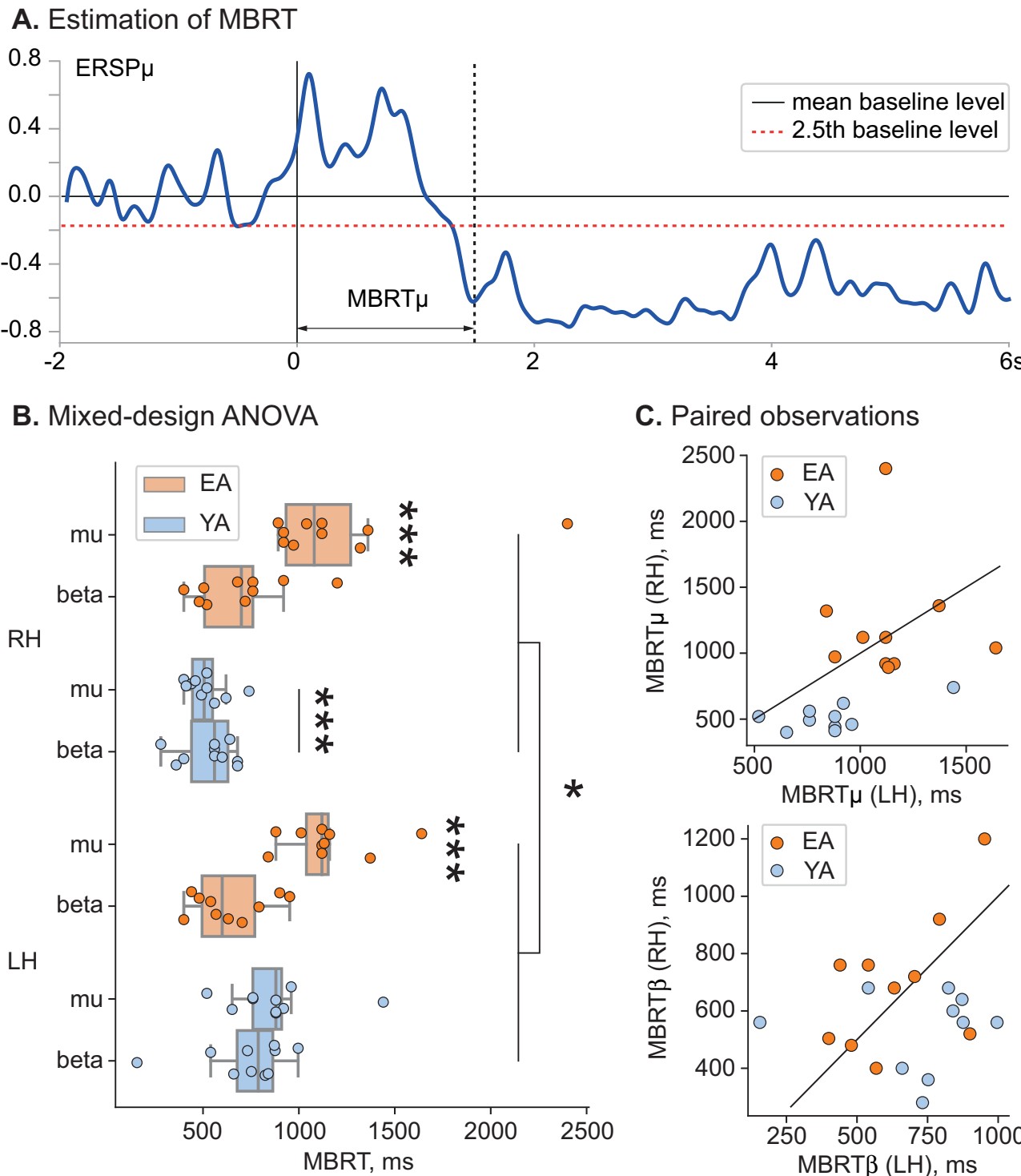

**Fig 2. Motor brain response time. A** An exemplary illustration of the $MBRT_\mu$ estimation. The blue curve shows single-subject $ERSP_\mu$ at the C4 sensor averaged over 15 LH epochs. Black solid and red dashed horizontal lines indicate mean and $2.5^{th}$ percentile level of the baseline $ERSP_\mu$, respectively. Black solid and black dashed vertical lines show the beginning of the audio command and estimated motor brain response, respectively. **B** Distribution of $MBRT_{\mu,\beta}$ across subjects in each (group,condition)-set. Here, '*' indicates $p < 0.05$ and '***' indicates $p < 0.001$. **C** Scatterplots of paired observations: (top) $MBRT_\mu(RH)$ versus $MBRT_\mu(LH)$ and (bottom) $MBRT_\beta(RH)$ versus $MBRT_\beta(LH)$ for each subject. Here, the diagonal line is $MBRT_{\mu,\beta}(RH) = MBRT_{\mu,\beta}(LH)$.

*PLI* is also formulated in the frequency domain. In this case, the definition of PLI given in Eq (2) is rewritten as:

$$\mathrm{PLI}_{i,j} = |\langle \mathrm{sign}(\mathrm{Im}[S_{i,j}(f)]) \rangle|, \tag{2}$$

where $S_{i,j}$ is a complex-valued Fourier-based cross-spectrum of $i^{th}$ and $j^{th}$ time-series and $f$ covers the frequency band of interest. Frequency-domain definition of PLI is implemented in MNE package and has been used in this study to reveal motor-related functional connectivity.

**Adjacency matrix.** The functional connectivity structure in each of the frequency bands of interest was presented by symmetric adjacency matrix sized (31 × 31). For each of $n = 20$ participants we calculated $k = 15$ connectivity matrices in both experimental conditions (LH and RH) during the premotor interval 0÷1.25 s contrasted by the baseline connectivity (−1.25÷0 s). Baseline contrast was applied to exclude false links, which could potentially arise due to the age-related changes in the resting-state functional connectivity. Then, for each subject, we computed mean connectivity matrices, averaged over $k = 15$ matrices, in both experimental conditions. To highlight statistically significant changes in functional connectivity related to the factor of age, we provided a between-subject analysis of the mean functional connectivity matrices. To address this issue we performed element-wise comparison of mean connectivity matrices for each type of movements between age groups using one-tailed unpaired $t$-test with $p_{pairwise} = 0.025$ ($dF = 9$, $t_{critical} = \pm 2.262$). Multiple comparison problem (MCP) was addressed via Network-Based Statistic (NBS) approach with $r = 2000$ random permutations and $p_{cluster} = 0.05$ [46].

## Results

### Motor brain response time analysis

First, we evaluated the effect of aging on the MBRT, i.e., the duration of the time interval required for the brain to activate a corresponding motor area for both groups. We estimated MBRT for each subject in mu and beta bands in both experimental conditions (Fig 2B) and compared the results taking into account Age, Movement Type and Frequency Band factors together (see Tables 1, 2 and 3). The mixed-design ANOVA test revealed a significant effect of Age ($F(1, 18) = 22.793$, $p < 0.001$), Band ($F(1, 18) = 19.226$, $p < 0.001$) and Movement Type ($F(1, 18) = 4.752$, $p = 0.043$) on MBRT. Post-hoc comparison via unpaired $t$-test indicated that the mean MBRT in EA group (M = 0.932, SD = 0.384) was significantly higher than mean MBRT in YA group (M = 0.66, SD = 0.234). Regarding the Frequency Band, post-hoc comparison via paired $t$-test demonstrated that the mean mu-band MBRT (M = 0.932, SD = 0.388) was significantly higher than mean beta-band MBRT (M = 0.648, SD = 0.211). Finally, post-hoc comparison via paired $t$-test showed that the mean MBRT in LH condition (M = 0.843, SD = 0.29) was significantly higher than mean MBRT in RH condition (M = 0.737, SD = 0.383).

**Table 1. Motor brain response time, s (Two-way mixed-design ANOVA summary).**

| Cases | *dF*1 | *dF*2 | Mean Square | F | p |
|---|---|---|---|---|---|
| Age (between-subject) | 1 | 18 | 1.359 | 22.793 | <.001*** |
| Band (within-subject) | 1 | 18 | 1.612 | 19.226 | <.001*** |
| Band * Age | 1 | 18 | 0.981 | 11.703 | 0.003** |
| Movement Type (within-subject) | 1 | 18 | 0.222 | 4.752 | 0.043* |
| Movement Type * Age | 1 | 18 | 0.548 | 11.739 | 0.003** |
| Band * Movement Type | 1 | 18 | 0.025 | 0.446 | 0.513 |
| Band * Movement Type * Age | 1 | 18 | 0.036 | 0.627 | 0.439 |

**Table 2. Motor brain response time, s (Post hoc comparisons Band—Age).**

|  |  | Mean Difference | SE | t | $p_{holm}$ |
|---|---|---|---|---|---|
| EA, mu | YA, mu | 0.482 | 0.085 | 5.693 | <.001*** |
|  | EA, beta | 0.505 | 0.092 | 5.519 | <.001*** |
|  | YA, beta | 0.545 | 0.085 | 6.430 | <.001*** |
| YA, mu | EA, beta | 0.023 | 0.085 | 0.274 | 1.000 |
|  | YA, beta | 0.062 | 0.092 | 0.681 | 1.000 |
| EA, beta | YA, beta | 0.039 | 0.085 | 0.463 | 1.000 |

**Table 3. Motor brain response time, s (Post hoc comparisons Movement Type—Age).**

|  |  | Mean Difference | SE | t | $p_{holm}$ |
|---|---|---|---|---|---|
| EA, RH | YA, RH | 0.426 | 0.073 | 5.846 | <.001*** |
|  | EA, LH | 0.060 | 0.068 | 0.881 | 0.400 |
|  | YA, LH | 0.155 | 0.073 | 2.132 | 0.120 |
| YA, RH | EA, LH | -0.366 | 0.073 | -5.020 | <.001*** |
|  | YA, LH | -0.271 | 0.068 | -3.964 | 0.004** |
| EA, LH | YA, LH | 0.095 | 0.073 | 1.306 | 0.400 |

Moreover, there was a significant interaction between the Band and Age of the participants ($F(1, 18) = 11.703$, $p = 0.003$). We could interpret this interaction as meaning that the different frequency bands activated differently in EA and YA groups. Particularly, mu-band MBRT w shown to be higher in EA group (EA, mu-band: M = 1.173, SD = 0.35) compared with the other (group,band)-pairs: (EA, beta-band: M = 0.668, SD = 0.213; YA, mu-band: M = 0.69, SD = 0.255; EA, beta-band: M = 0.628, SD = 0.213).

Finally, there was a significant interaction between the Movement Type and Age of the participants ($F(1, 18) = 11.739$, $p = 0.003$). We could interpret this interaction as meaning that the Movement Type influenced MBRT differently in EA and YA groups. Particularly, YA group reacted significantly faster in RH condition (YA, RH: M = 0.524, SD = 0.119) compared with the other (group,condition)-pairs: (YA, LH: M = 0.795, SD = 0.244; EA, RH: M = 0.95, SD = 0.44; EA, RH: M = 0.89, SD = 0.328). According to the results of paired observation (Fig 2C), 9 of 10 subjects in YA group demonstrated that $MBRT_{\mu(LH)>}MBRT_{\mu(RH)}$ and 4 of 10 subjects in EA group had the same effect. Regarding the estimations of $MBRT_{\beta}$, 8 of 10 subjects in YA group showed that $MBRT_{\beta(LH)>}MBRT_{\beta(RH)}$, while in EA group the same effect was demonstrated in 2 of 10 subjects.

## Within-subject time-frequency analysis

Based on the above MBRT analysis, we assumed that age-related changes affecting the speed of brain motor activation should be found in the motor initiation period. With this aim, we performed within-subject spatio-temporal clustering analysis of the spectral power in the theta, alpha/mu and beta frequency bands for each (group, condition)-set during the motor initiation (0÷1.5 s). Fig 3 shows the results of within-subject clustering analysis in the LH condition for both groups of subjects. It is seen that in the LH condition (non-dominant hand movement), brain activation in both YA and EA groups proceeds similarly. Specifically, the suppression of beta-rhythm in the motor cortex at 472 ms (YA group) and 308 ms (EA group) was followed by the mu-band ERD at 604 ms (YA group) and 548 ms (EA group) and was related to the motor execution control. Desynchronization of the beta and mu oscillations was

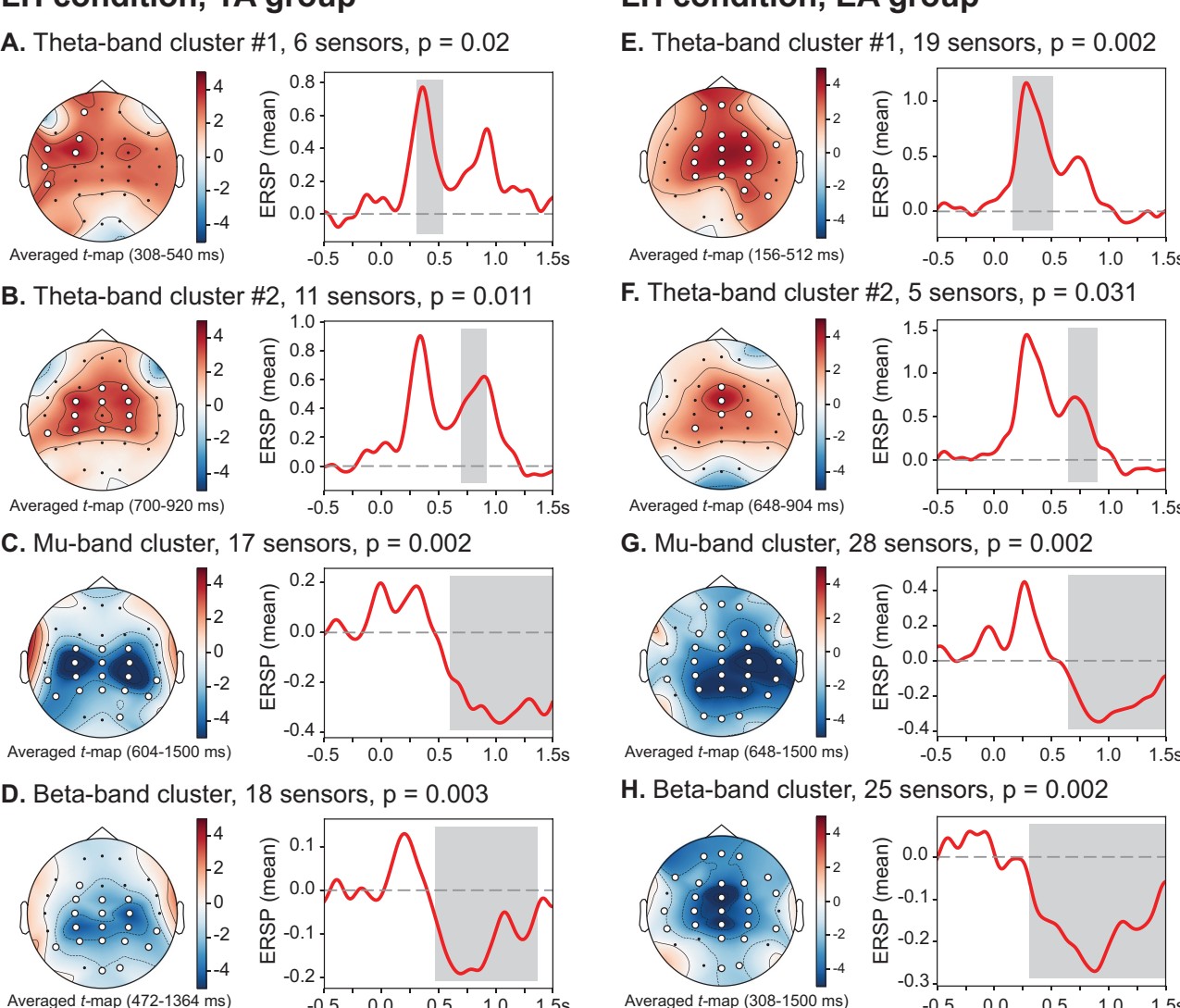

**Fig 3. Sensor-level within-subject time-frequency analyses during motor initiation (LH condition).** Baseline-corrected spatio-temporal clusters (left) and mean ERSP of the corresponding clusters (right): (**A, B, E, F**) theta clusters; (**C, G**) alpha/mu clusters; (**D, H**) beta clusters. White dots indicate sensors exhibiting significant spectral power changes. Pairwise comparison is performed via one-tailed one-sampled $t$-test with $p_{pairwise}$ = 0.005 ($dF$ = 9, $t_{critical}$ = ±3.2498) and cluster-based analysis is performed via non-parametric permutation test with $p_{cluster}$ = 0.05.

preceded by the theta-band activation from 308 to 540 ms (YA group) and from 156 to 512 ms (EA group). In the YA group, this theta-band cluster involved the left frontal (Fp1, F3), frontal-central (FC3) and temporal (FT7, T7, TP7) EEG sensors. In the EA group, strong theta-band synchronization spanned widely across the frontal, central and occipito-parietal EEG sensors. Also, a spatio-temporal cluster showing a significant activation in theta band appeared almost simultaneously with a mu-band desynchronization: from 720 to 920 ms (YA group) and from 648 to 904 ms (EA group). Here, a significant theta-band activation was shown in frontal (Fz, F4), central (FC3, FCz, FC4, C3, C4, CP3, CPz, Cp4) and left temporal (TP7) EEG sensors in YA group, while in EA group midline (Fz, Fcz, Cz) and bilateral central (C4, CP3) EEG sensors indicated increased theta-band ERSP. Thus, in LH condition, both groups shared a similar activation mechanism and timing of the motor initiation process.

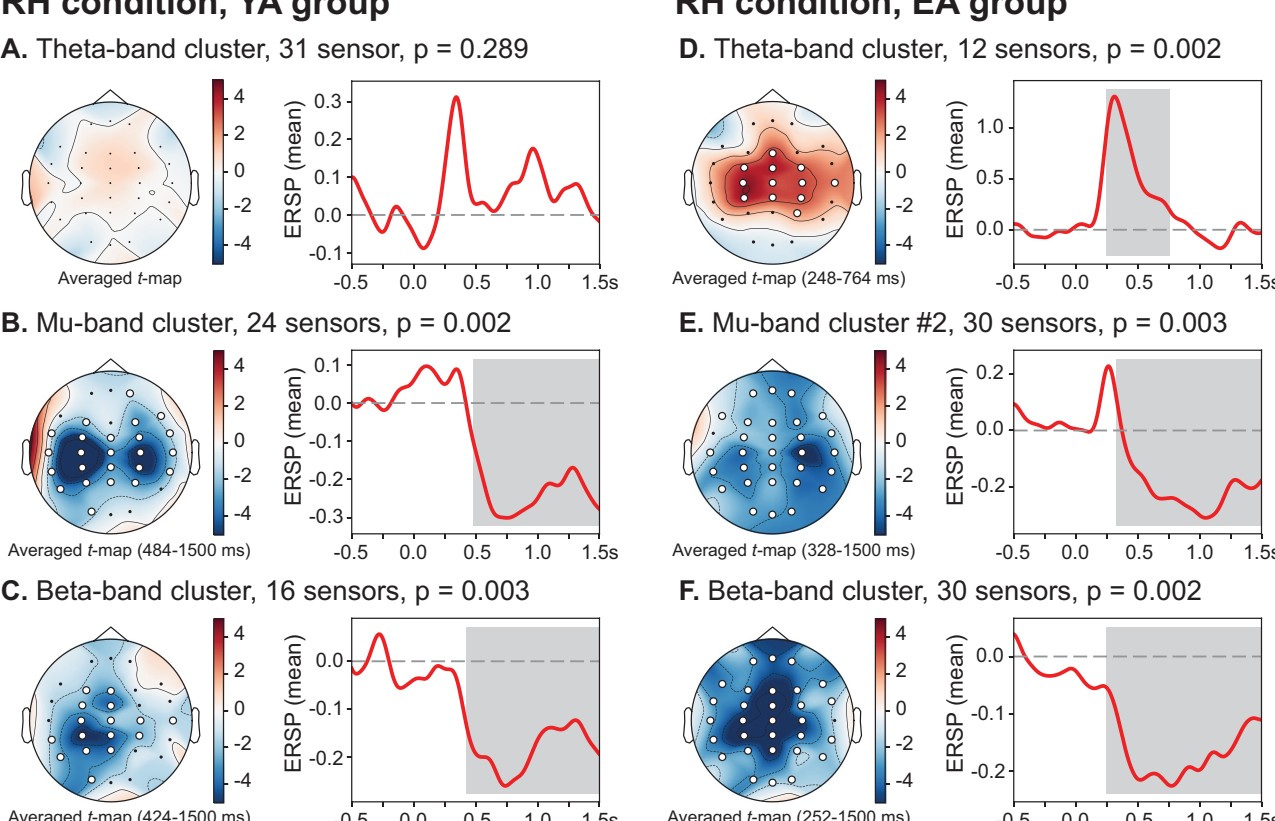

**Fig 4. Sensor-level within-subject time-frequency analyses during motor initiation (RH condition).** Baseline-corrected spatio-temporal clusters (left) and mean ERSP of the corresponding clusters (right): (**A, D**) theta clusters; (**B, E**) alpha/mu clusters; (**C, F**) beta clusters. White dots indicate sensors exhibiting significant spectral power changes. Pairwise comparison is performed via one-tailed one-sampled $t$-test with $p_{pairwise} = 0.005$ ($dF = 9$, $t_{critical} = \pm 3.2498$) and cluster-based analysis is performed via non-parametric permutation test with $p_{cluster} = 0.05$.

On the contrary, the way of cortical activation during the motor initiation period in the RH condition (dominant hand movement) was different in considered age groups (Fig 4). In both groups, beta- and mu-band ERD in RH condition started earlier compared with LH condition: from 424 ms (YA group) and 252 ms (EA group) for beta-band ERD; from 484 ms (YA group) and 328 ms (EA group) for mu-band ERD. However, in the YA group, the theta-band spectral power did not change significantly during the pre-movement period. At the same time, the theta-band activation in the RH condition similar to LH condition was observed in the EA group (248-746 ms) involving frontal (Fz), central (FC line, C line, CP line), parietal (P4) and right temporal (T8) EEG sensors.

## Between-subject time-frequency analysis

To address the age-related changes in the pre-movement theta-band activation in detail, we provided a between-subject spatio-temporal clustering analysis of ERSP separately in each experimental condition. In LH condition, the significant between-subject difference in the theta-band activation was not observed. On the contrary, the between-subject differences were found in RH condition from 364 to 512 ms: the theta-band spatio-temporal cluster included C3, C4, Cp3, Cpz, Cp4 and P4 EEG sensors (Fig 5A).

## RH condition, between-subject analysis (EA-YA)

**A.** Theta cluster, 6 sensors, p = 0.049

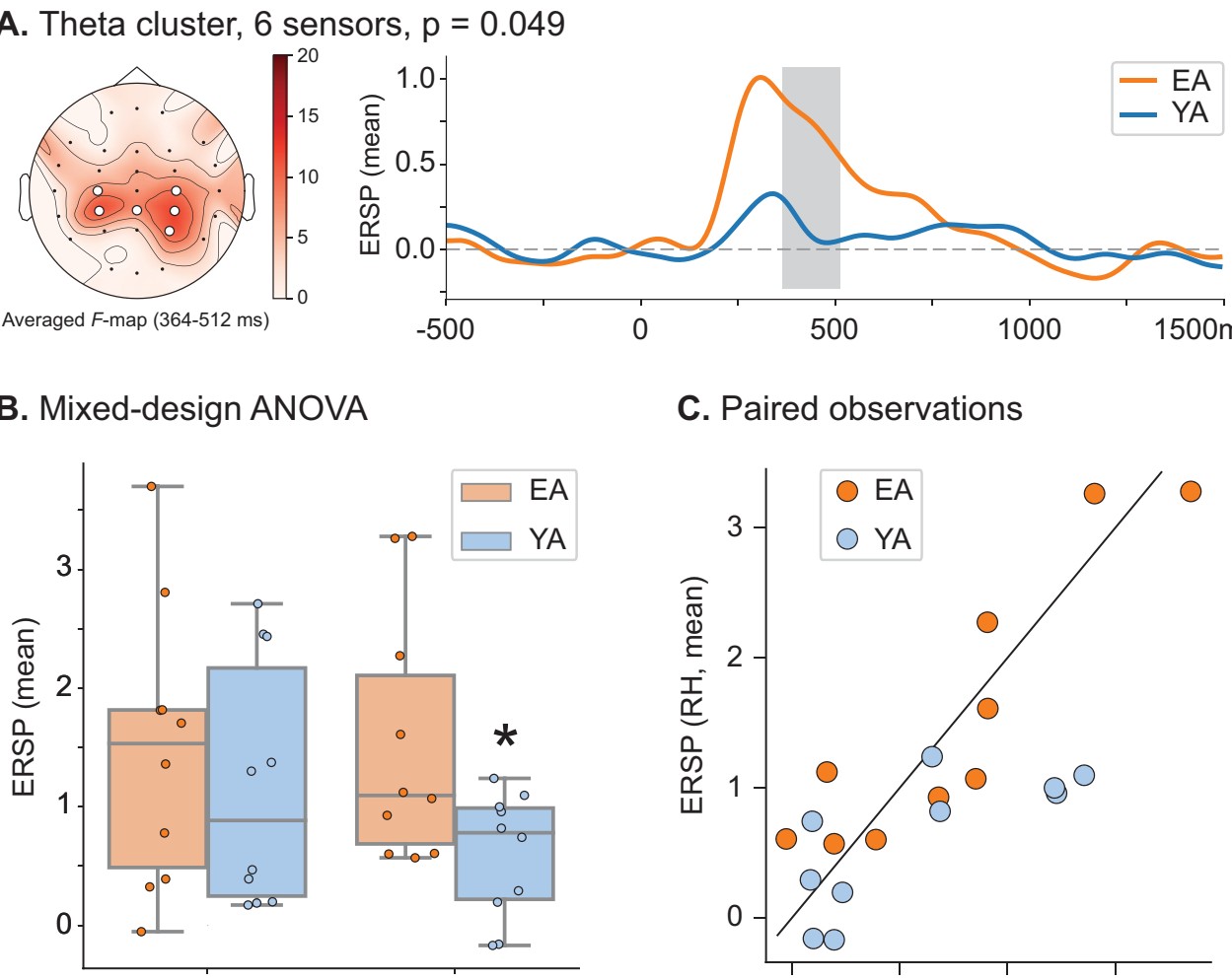

**Fig 5. Between-subject analyses of the theta-band activation during motor initiation. A** Baseline-corrected spatio-temporal clusters (left) and mean theta-band ERSP of the corresponding clusters (right) preceding RH movements execution. White circles indicate cluster of sensors with significant differences via non-parametric test. Pairwise comparison is performed via one-tailed unpaired $F$-test with $p_{pairwise}$ = 0.005 ($dF1$ = 1 and $dF2$ = 18, $F_{critical}$ = 10.218) and cluster-based analysis is performed via non-parametric permutation test with $p_{cluster}$ = 0.05. **B** Distribution of the event-related theta-band spectral power of the uncovered spatio-temporal cluster (**A**) across subjects in each (group, condition)–set. Here, '\*' indicates $p < 0.05$. **C** Scatterplot of paired observations within each group.

To estimate age-related differences in theta-band activation taking into account both Age and Movement Type factors, we compared mean theta-band spectral power over the evaluated spatio-temporal cluster via mixed-designed ANOVA (Fig 5B and the summary is presented in detail in Tables 4 and 5). The mixed-design ANOVA test revealed no significant effect of both Age ($F(1, 18)$ = 2.189, $p$ = 0.156) and Movement Type ($F(1, 18)$ = 3.151, $p$ = 0.093) on the pre-movement theta-band spectral power. However, there was a significant interaction between these factors ($F(1, 18)$ = 5.085, $p$ = 0.037). We could interpret this interaction as follows: pre-movement theta-band power was similar for LH condition the EA and YA groups (EA, LH: M = 1.464, SD = 1.171; YA, LH: M = 1.169, SD = 1.038), while the YA group demonstrated

**Table 4. Pre-movement theta-band spectral power (Two-way mixed-design ANOVA summary).**

| Cases | dF1 | dF2 | Mean Square | F | p |
|---|---|---|---|---|---|
| Age (between-subject) | 1 | 18 | 3.757 | 2.189 | 0.156 |
| Movement Type (within-subject) | 1 | 18 | 0.626 | 3.151 | 0.093 |
| Movement Type * Age | 1 | 18 | 1.010 | 5.085 | 0.037* |

**Table 5. Pre-movement theta-band spectral power (Post hoc comparisons—Age-Movement Type).**

| | | Mean Difference | SE | stat | p |
|---|---|---|---|---|---|
| EA, LH | YA, LH | 0.295 | 0.438 | 42.0 (U) | 0.28 |
| | EA, RH | -0.068 | 0.199 | 21.0 (W) | 0.51 |
| | YA, RH | 0.863 | 0.438 | 2.503 (t) | 0.022* |
| YA, LH | EA, RH | -0.363 | 0.438 | 25.0 (U) | 0.032* |
| | YA, RH | 0.568 | 0.199 | 7.0 (W) | 0.037* |
| EA, RH | YA, RH | 0.931 | 0.438 | 25.0 (U) | 0.032* |

*According to the results of Shapiro-Wilk normality test, the (EA,RH)-set with p = 0.029 and (YA,LH)-set with p = 0.034 did not come from the normally distributed population. Therefore, along with the unpaired t-test (t) we used non-parametric Mann-Whitney U-test (U) and Wilcoxon signed-rank test (W) to the provide the post hoc comparisons.*

lower pre-movement theta-band power in RH condition (EA, RH: M = 1.532, SD = 1.054; YA, RH: M = 0.601, SD = 0.520). According to the paired observations (Fig 5C), 8 of 10 subjects in YA group demonstrate the effect.

## Functional connectivity analysis

To support and extend our observations of the cortical activation during motor initiation, we explored age-related changes in terms of the underlying functional interactions between remote brain regions. Due to previously uncovered between-subject difference in the theta-band activity, we provided a between-subject comparison of the sensor-level theta-band functional connectivity estimated during the pre-movement stage in the RH condition (0÷1.25 s). As seen in Fig 6A, the distributed functional network with strong hubs in occipito-parietal (O1, O2, P3, P7), frontal (F7) and midline (Oz, Pz, CPz, FCz) EEG sensors was highly coupled in YA group compared to EA subjects. At the same time, we found the significant bilateral coupling increase between central (Cz, C3, C4, Cp3, Fc4), temporal (TP7, TP8, T7, FT7), and frontal (Fp1, Fp2, F3, F4) EEG sensors in EA participants (Fig 6B). Here, Cz sensor being a strong hub of the functional network provided a large-scale neuronal communication via coupling with the bilateral cortical sensorimotor circuits (C3–TP7 and C4–TP8), along with temporal (Cz–FT7, Cz–T7) and frontal (Cz–F3, Cz–F4, Cz–Fc4) EEG sensors.

## Discussion

We considered the effect of healthy aging on the cortical activation in the motor initiation phase during the controlled repetition of fine motor tasks—squeezing one of the hands into a fist paced by the audio command. We found that the time required for motor-related mu- and beta-band desynchronization, which we referred to as a motor brain response time (MBRT), was increased in the elderly subjects compared to the younger control group during the dominant hand task. Based on the results of time-frequency and functional connectivity analyses,

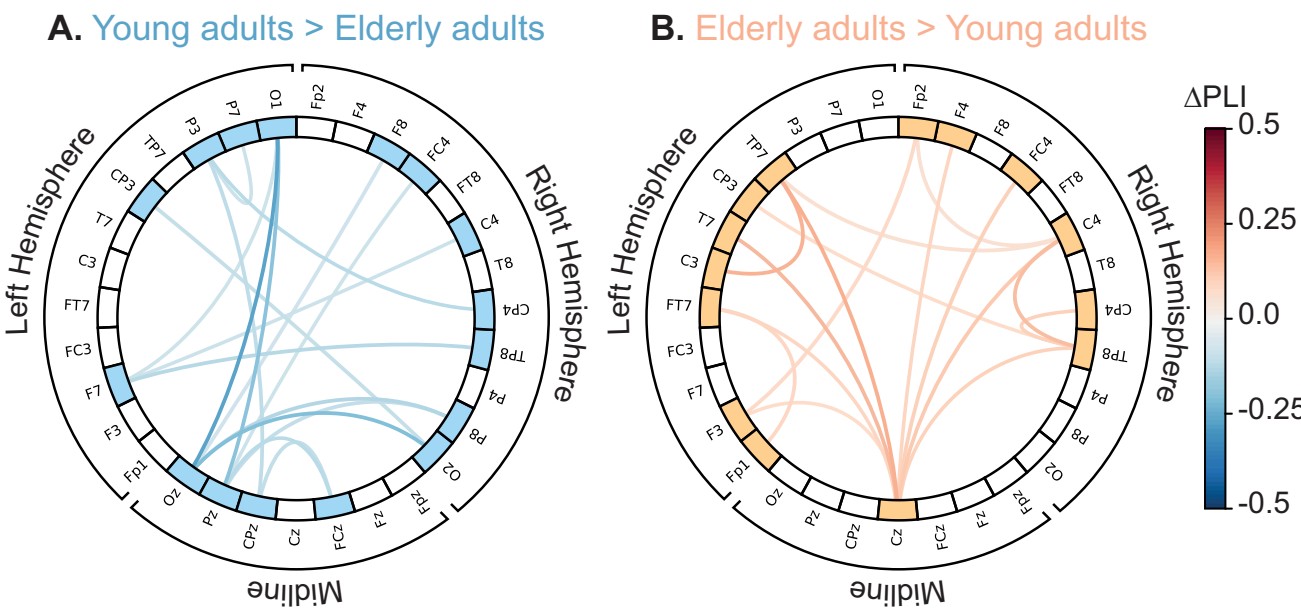

**Fig 6. Between-subject analysis of the theta-band functional connectivity during motor initiation in the RH condition. A** Significantly stronger coupling in YA compared to the EA. **B** Significantly stronger coupling in EA compared to the YA. Here, ΔPLI defines the difference between group-level mean functional connectivity (EA versus YA). Element-wise comparison of mean connectivity matrices between Age groups was performed via one-tailed unpaired $t$-test with $p_{pairwise} = 0.025$ ($dF = 9$, $t_{critical} = \pm2.262$).

we found that the prolonged motor response was preceded by the increased theta-band activation in central, central-parietal, and parietal EEG sensors, along with a stronger coupling between central, bilateral temporal, and frontal sensors. Further, we discuss our results in the context of possible mechanisms supporting the motor initiation slowdown.

We observed the significant age-related differences in the MBRTs, which demonstrated a higher speed of the motor initiation in the case of the dominant (right) hand task in younger participants compared to the elderly adults. At the same time, motor initiation was equally slow during the non-dominant (left) hand task in both age groups. Moreover, MBRTs of elderly adults in both conditions approached the level of the non-dominant hand in younger subjects. Based on these findings, we suggested that the neuronal mechanisms supporting right-hand dominance are impaired under healthy aging. Despite the conflicting evidence in the literature, our results are consistent with several studies showing a similar effect. First, T. Kalisch et al. [8] demonstrated the behavioral decline in the dominant hand performance leading to ambidexterity in elderly adults. The authors argued that their findings could be explained by the mechanism of use-dependent plasticity [22], causing the degradation of well-trained motor functions due to the reduced activity and sedentary lifestyle of elderly individuals. Also, J. Langan et al. [17] supported these results and showed less-lateralized task-related motor activity in elderly adults compared to the younger control group. They found that longer reaction time in elderly adults was correlated with greater activation of the ipsilateral primary motor cortex during the motor task performance and weaker resting-state interhemispheric coupling, which was also observed in Refs. [16, 47, 48]. Described changes provided the compensatory mechanism to maintain the level of motor performance consisting in the reorganization of functional networks aimed at overcoming the age-related chemical

and structural changes [20, 21]. Our results also evidence the motor-related over-activation of the brain areas in elderly adults as a large cluster of mu- and beta-band desynchronization covering additional frontal, central, parietal, occipital, and temporal EEG sensors (see Figs 3 and 4).

However, the aforementioned mechanisms are not the only ones that support the brain's motor response slowdown. Our results showed that the prevalent theta-band activation in central, central-parietal, and parietal EEG sensors preceded the motor-related mu-band desynchronization during the non-dominant hand movements in both groups and the dominant hand movements in elderly adults. The mechanism of motor initiation related to the increased theta activity is explained by the Bland's sensorimotor integration model. In their early works on rodents [49, 50], B.H. Bland with colleagues treated the hippocampal formation theta activity as a communication channel between the sensory processing and movement initiation. Further, the Bland's model was extended to the human brain in a series of works by J.B. Caplan et al. [25, 51]. In their studies, they concluded that while mu-band suppression (a traditional hallmark of the motor-related brain activity) reflected cortical activation directly during the motor task execution, the increased theta power between stimuli presentation and motor execution was associated with sensorimotor integration similarly to rodents. Along with this, several EEG-studies reported the increase of theta-band power during the planning phase in the choice-related, catching, and imagery motor tasks [52–54]. Specifically, M. Tambini et al. [53] demonstrated a positive correlation between theta-power and task performance. On the contrary, we found that increased theta-band power was associated with prolonged motor initiation. It should be noted that the significant increase of the theta-band power related to the dominant hand decline in elderly adults was observed in the central, central-parietal, and parietal EEG sensors covering the sensorimotor area. Following the recent study by J. Dushanova et al. [55], such result should be explained by the different strategies of the motor task initiation between age groups. While the degraded plasticity in elderly adults requires higher cortical activation for motor planning, younger subjects optimize their cognitive resources for the familiar and well-trained motor task accomplishment. The latter was represented as a lower theta-band activation. Therefore, less effective use of cognitive resources slowed the motor planning phase in elderly adults compared to the younger control group during the dominant hand tasks.

These conclusions were also supported and extended by the results of functional connectivity analysis during the pre-movement phase. The differences in theta-band functional connectivity between two groups could be interpreted as a meaning of the different mechanisms of cortical interaction that subserve motor planning in elderly adults and young subjects. First, we showed that in young adults, pre-movement theta-band functional connectivity strongly involves midline EEG sensors. According to the previous studies [56–59], strong midline coupling could be interpreted as increased perceptual-motor facility and motor working memory. Thus, we suppose that in young adults, initiation of the familiar motor activity emphasizes motor working memory and enables the formation and processing of the motor memories, i.e., the stored information about the motor action obtained from prior experience, for accurate motor performance [60]. On the contrary, in elderly adults, we observed a completely different structure of the sensor-level functional connectivity, i.e., a stronger coupling between the frontal, central-parietal, and bilateral temporal EEG sensors with the most influential node located in the central EEG row (Cz sensor). As the working memory decline with age is well-documented [61–63], we conclude that memory representation of motor actions is less accessible in elderly adults. Based on our findings and the existing literature, we suggest that higher coupling within the sensorimotor area during a pre-movement phase in the elderly group indicates the prevalence of sensorimotor integration mechanisms relative to the resource-

demanding Bland's Type 1 motor-related theta activation [25]. We conclude that the uncovered differences in the cortical activation, related with an increased theta-band power, taken together with age-related changes in neural interactions reflect non-optimal utilization of the cognitive brain resources in elderly adults causing the significantly delayed motor initiation process.

## Conclusion

Elderly adults exhibited the approach to ambidexterity in term of the slowdown in cortical activation related to the execution of the dominant hand task. We showed that motor-related mu- and beta-band desynchronization appeared faster in young subjects during dominant hand movement, while in elderly adults it appeared equally slow in both hands. We demonstrated that the observed age-related loss of the dominant hand advance was accompanied by the increased theta-band activation similar to Bland's Type 1 sensorimotor integration model. At the same time, age-related changes affected the structure of sensor-level functional connectivity during motor initiation: younger subjects demonstrated stronger interaction between frontal, parietal, and midline EEG sensors, while elderly adults demonstrated higher coupling between central, temporal, frontal sensors. Taken together, our results on cortical activation and underlying neuronal interactions suggest the utilization of more demanding pre-movement processes in elderly adults causing a significant slowing down in the motor initiation.

## Acknowledgments

The authors gratefully acknowledge the anonymous reviewer for a detailed feedback and valuable comments.

## Author Contributions

**Conceptualization:** Nikita S. Frolov, Vladimir A. Maksimenko, Alexander E. Hramov.

**Data curation:** Vadim V. Grubov, Anton R. Kiselev.

**Formal analysis:** Nikita S. Frolov, Elena N. Pitsik, Vladimir A. Maksimenko, Zhen Wang.

**Funding acquisition:** Nikita S. Frolov, Alexander E. Hramov.

**Investigation:** Nikita S. Frolov, Elena N. Pitsik.

**Methodology:** Nikita S. Frolov.

**Project administration:** Nikita S. Frolov, Alexander E. Hramov.

**Software:** Nikita S. Frolov, Elena N. Pitsik.

**Supervision:** Nikita S. Frolov, Alexander E. Hramov.

**Visualization:** Nikita S. Frolov.

**Writing – original draft:** Nikita S. Frolov, Elena N. Pitsik, Alexander E. Hramov.

**Writing – review & editing:** Nikita S. Frolov, Vladimir A. Maksimenko, Alexander E. Hramov.

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
