## [Decision Letter · Decision Letter 0]

25 Jun 2020

PONE-D-20-14466

Age-related changes in the motor planning strategy slow down motor initiation in elderly adults

PLOS ONE

Dear Dr. Frolov,

Thank you for submitting your manuscript to PLOS ONE. After careful consideration, we feel that it has merit but does not fully meet PLOS ONE’s publication criteria as it currently stands. Therefore, we invite you to submit a revised version of the manuscript that addresses the points raised during the review process.

We look forward to receiving your revised manuscript.

Kind regards,

Mukesh Dhamala, Ph. D.

Academic Editor

PLOS ONE

Reviewers' comments:

Reviewer's Responses to Questions

**Comments to the Author**

1. Is the manuscript technically sound, and do the data support the conclusions?

Reviewer #1: Partly

2. Has the statistical analysis been performed appropriately and rigorously? 

Reviewer #1: Yes

3. Have the authors made all data underlying the findings in their manuscript fully available?

Reviewer #1: No

4. Is the manuscript presented in an intelligible fashion and written in standard English?

Reviewer #1: Yes

5. Review Comments to the Author

Reviewer #1: This is a particularly well designed and interesting study by Frolov and colleagues to understand the age associated changes in the motor initiation. However, please find below my general criticisms and suggestions about your approach to elucidate the mechanisms underlying age related differences in motor initiation.

Without digressing much, I do not necessarily agree that you could call any of findings of this study as potential mechanisms of motor planning and execution. Neuroscience has long been hailing some critical observations and phenomenon as mechanisms. I don’t cast necessarily a doubt on the overall findings and observations based on this study but hesitant to infer them as underlying mechanisms of motor planning and control during aging.

The findings in this paper at best in my opinion what we should call as empirical observations grounded on certain hypothesis backed up by extant literature on Cognitive Neuroscience of healthy and pathological Aging such as Compensation related Utilization of Resources, Dedifferentiation etc. applied for interpreting observations during finer motor task performances. On the same token, Mu band ERD signatures and functional connectivity based on PLI is not the mechanisms but qualitative and quantitative observations associated with the aging process. May I also ask you what exactly is the underlying mechanism associated with aging that causally explain these observations? I don’t think a clear answer would emerge unless you do some computational modelling or DCM etc. Since, You have a clear hypothesis why didn’t you think of it on the first place to do a systematic model validation based on your EEG observations. I would suggest that this would be of great interest to test out the alternate hypothesis and mechanisms to explain these observations based on change in ambidexterity with aging in the elderly group. Yes agreed that this may be related to slowing down and also recruitment of large scale brain networks as we know there is more integration with aging at the expense of specialization. But then again, these competitive mechanisms has not really been tested in my opinion.

Having said this, I do think these observations are necessary in understanding the motor planning and control strategies with healthy and pathological aging process. In this regard, authors have made some interesting observations similar to what is somewhat known in the existing literature.

They found that the motor cortex of younger adults activated much faster during the dominant hand task compared to elderly adults and the time required for motor activation in elderly group was equal for both hands and approached the level of the non-dominant hand of younger adults. Furthermore, they found significant differences in cortical activation during the time interval preceding the motor action (during motor initiation phase). In elderly adults, as well as in young adults performing the non-dominant hand task, they observed the increased theta-band power in sensorimotor and frontal areas, whereas theta-activation was insignificant in young adults during the dominant hand task. Finally, based on the results of between-subject functional connectivity analysis, they revealed that motor planning involves different types of cortical interactions in young adults and elderly adults, which allows concluding about age-related changes in motor planning mechanisms.

The article is very well written with no ambiguity in understanding the key message and the narrative. The materials & methods are also clearly presented. The introduction and discussion carefully construed backed up by appropriate and relevant literature review and summary. Below I outline my suggestions, questions and corrections for all the sections.

Abstract and title

Perhaps it would be better if you change the article title and the abstract a bit. My reading of your article gives me an impression that this article mainly concerns about movement initiation phase rather than the entire motor planning strategy and contingency. You are interested in understanding behavioural characteristics RT, Response Accuracy etc. is affected due to differences ensuing in the motor initiation phase. In my opinion, a more appropriate title replacement would be “Age-related slowing down in the motor initiation in elderly adults”.

Materials and methods

In methods section, you talk about background recording before recording active phase. You could simply say 5 minutes Eyes Open Resting state. They don’t appear to be any different to me. In the active phase you say you have 60 fine motor tasks per participant and 30 tasks per hand. The duration of the beep short or long provides cue which hand (dominant vs. non-dominant) to use. This is fine, but how different are these 60 tasks actually from each other. Are they really all different or similar? Could you provide a statistical summary or similarity measure to point out the differences between categories of finer motor task categories. If the task categories are dissimilar then the motor signal or changes in motor signals would be more enhanced irrespective of the age category. In this regard, just a clarification will suffice. I am a bit confused as the Experimental paradigm presented in figure 1 clearly shows a single motor task (I guess squeezing wrist of one hand with the other)

In line 124, authors suggest that a priory knowledge about the cortical activation during movements execution implies that motor brain response is determined as a pronounced event-related desynchronization (ERD) of mu-oscillations in the contralateral area of the motor cortex. Therefore, they recorded and analysed activity from symmetric sensors C3 and C4 respectively to record mu band response time (MBRT).

I am wondering based on the recent literature (which is by the way not referenced) Transient human movement is served by a specific pattern of neural oscillatory activity, particularly in the beta band (14–30 Hz). Briefly, prior to and during movement, there is a strong decrease in beta activity relative to baseline levels, known as the peri-movement beta event-related desynchronization (ERD), which begins about 1.0 s before movement onset and dissipates shortly after movement concludes. Actually, roughly 500 ms following stimulus offset or the completion of an actual voluntary or passive movement, or imagined movement, the beta rhythm increases in magnitude with respect to a baseline period preceding the event. This period of ERS is often termed the post-movement beta rebound (PMBR). Could you please comment why are you not looking at this Beta ERD. Is not that relevant? I guess post-movement you should also check PMBR for frequency and amplitude. May be something interesting out there.

Based on my understanding of the aging literature on motor signals, older adults exhibited an almost threefold increase in spontaneous beta power in the primary motor cortices, as well as significantly stronger beta ERD in the same regions compared to younger adults. Furthermore, it has been shown that during simple movements, these beta-band oscillations reliably peak in the precentral gyri bilaterally with stronger activity contralateral to movement, while more complex movements (and some simple movements) also induce activity in the supplementary motor area and bilateral premotor cortices, postcentral gyri, parietal cortices, and cerebellum. Perhaps, it would be necessary to discuss about this in the intro and discussion section.

Results

On the contrary, the between-subject differences were found in the spatial cluster, which included Cp3, Cpz, and Cp4 sensors (dorsal stream region of the sensorimotor area). Did you carry out a source analysis or else how are you actually talking about Ventral and Dorsal regions? I thought you have restricted your analysis at the sensor level. I wouldn’t talk about network based on sensor based functional connectivity in the time or frequency domain. Spatial resolution is too poor to argue for this even if you observed spectro-temporal signatures. For example, you mention distributed frontoparietal network. If I ask you how many areas do you think FPN comprise of then how would you answer? Please stick to sensor level analysis of your brain signals and connectivity patterns among sensor groups to discern changes. I don’t agree that talking about brain areas, ventral and dorsal streams etc. and above all networks makes much sense. You would probably admit networks in this case is loosely defined. This is also not a MEG Study.

Discussion

I enjoyed reading the discussion section. Having said that I think the authors need to seriously look at some of their assertions. One thing is to see a change and other is to speculate about those change. Many of the formulations and interpretations are at this point remain unverified and speculative. Not a clear demonstration yet. However the good thing is that the authors have cited and covered references which are most relevant to their findings based on mainly two types of analysis time-frequency and functional connectivity. Further, they discussed various mechanisms which render support to their empirical observations and relates to existing literature. In particular, this is one of the key reason why instead of listing a series of plausible mechanisms, the authors could identify and evaluate quantitatively one or two key mechanisms highlighted in the discussion section. For example, the observation of increase in theta band activity/power in the Frontal sensors and in the sensors near sensorimotor areas could be sensorimotor integration (provided by Bland’s model) mechanisms or this could be long range connectivity change between distal brain areas leading to an elevation in the theta band power. Also, I don’t like the fact that you keep mentioning throughout the article about brain areas while you actually carry out all of your analysis primarily at the EEG sensor level. They clearly do not commensurate with each other. Please, see my previous comments on this issue persistent in this manuscript.

Please see below an example of what I mean.

You write in line number 340-342 “that It should be noted that the significant increase of the theta-band power related to the dominant hand decline in elderly adults was observed in the dorsal stream region of the sensorimotor cortex and associated with the motor 342 planning but not with audio command processing”. How do you establish that? You don’t look at structure/anatomy neither you acquire MEG/fMRI data nor do any source estimation. Then I fail to understand what’s actually the basis for making the above statement. The above may be possible but to pinpoint that precisely you do need concurrent EEG-MRI evidence in my opinion.

Minor

Line number 190 may have a typo. Please check. Same pattern of typo recurring throughout the manuscript. Actually spotted them in several other places in this manuscript. Please revisit those sections where these typos are present and rectify.

6. PLOS authors have the option to publish the peer review history of their article (what does this mean?). If published, this will include your full peer review and any attached files.

Reviewer #1: No

---

## [Author Response · Author response to Decision Letter 0]

5 Aug 2020

Response to the Reviewer #1: First, we would like to thank the Reviewer for a careful reading of our paper, pointing the major and minor issues, and a detailed feedback, which has been extremely helpful for improving the quality of the revised manuscript. We are also delighted to know that the Reviewer has found our study interesting and well-designed. We have done our best to address all the issues raised by the Reviewer. Below, please find our response to the Reviewer’s comments.

Reviewer #1 Comments

Abstract and title

Comment: Perhaps it would be better if you change the article title and the abstract a bit. My reading of your article gives me an impression that this article mainly concerns about movement initiation phase rather than the entire motor planning strategy and contingency. You are interested in understanding behavioural characteristics RT, Response Accuracy etc. is affected due to differences ensuing in the motor initiation phase. In my opinion, a more appropriate title replacement would be “Age-related slowing down in the motor initiation in elderly adults”.

Answer: We thank the reviewer for the criticism. We found that the referee’s suggestions about the title fits better to the main results presented in our study, so we corrected it accordingly. We also modified the abstract putting more emphasis on our experimental observation than on our conclusions.

Materials and methods

Comment: In methods section, you talk about background recording before recording active phase. You could simply say 5 minutes Eyes Open Resting state. They don’t appear to be any different to me. In the active phase you say you have 60 fine motor tasks per participant and 30 tasks per hand. The duration of the beep short or long provides cue which hand (dominant vs. non-dominant) to use. This is fine, but how different are these 60 tasks actually from each other. Are they really all different or similar? Could you provide a statistical summary or similarity measure to point out the differences between categories of finer motor task categories. If the task categories are dissimilar then the motor signal or changes in motor signals would be more enhanced irrespective of the age category. In this regard, just a clarification will suffice. I am a bit confused as the Experimental paradigm presented in figure 1 clearly shows a single motor task (I guess squeezing wrist of one hand with the other)

Answer: We thank the reviewer for this comment. Indeed, the experimental task in the original manuscript was described in misleading form. We should have used a term ‘repetitions’ instead of ‘tasks’, since each participant was asked to perform multiple repetitions of the same fine motor task (squeezing a hand into a wrist after the audio signal and holding it until the second signal) using either left or right hand (30 repetitions per hand, 60 in total). We clarified this point in the revised version of the manuscript. We also corrected the description of the background recordings accordingly.

Comment: In line 124, authors suggest that a priory knowledge about the cortical activation during movements execution implies that motor brain response is determined as a pronounced event-related desynchronization (ERD) of mu-oscillations in the contralateral area of the motor cortex. Therefore, they recorded and analysed activity from symmetric sensors C3 and C4 respectively to record mu band response time (MBRT).

I am wondering based on the recent literature (which is by the way not referenced) Transient human movement is served by a specific pattern of neural oscillatory activity, particularly in the beta band (14–30 Hz). Briefly, prior to and during movement, there is a strong decrease in beta activity relative to baseline levels, known as the peri-movement beta event-related desynchronization (ERD), which begins about 1.0 s before movement onset and dissipates shortly after movement concludes. Actually, roughly 500 ms following stimulus offset or the completion of an actual voluntary or passive movement, or imagined movement, the beta rhythm increases in magnitude with respect to a baseline period preceding the event. This period of ERS is often termed the post-movement beta rebound (PMBR). Could you please comment why are you not looking at this Beta ERD. Is not that relevant? I guess post-movement you should also check PMBR for frequency and amplitude. May be something interesting out there.

Answer: We thank the reviewer for this valuable remark. Indeed, the level of neocortical beta-band oscillations is considered as relevant marker of declined motor performance in healthy ageing and disease. Also, a peri-movement beta-band ERD emerging slightly before the motor action is known to be associated with motor planning [Heinrichs-Graham E., et al. (2016). Journal of cognitive neuroscience]. According to the reviewer’s comment we have discussed this topic in the intro section.

We have also modified the analysis of MBRT by additional consideration of MBRT in the beta-band (see Fig.2 and subsection ‘Motor brain response time analysis’ in the Results section). Specifically, during this analysis we have found that beta-band MBRT reflects the same properties as a mu-band MBRT (the fastest brain response has been observed in RH condition in YA group). Also, we observed that beta-band MBRT is significantly lower than mu-band MBRT, that is consistent with the existing literature and reviewer’s comment.

We suppose, that the provided extended analysis of MBRT has gained the relevance of our conclusions.

We also thank the reviewer for an interesting idea’s for the continuation of a current research.

Comment: Based on my understanding of the aging literature on motor signals, older adults exhibited an almost threefold increase in spontaneous beta power in the primary motor cortices, as well as significantly stronger beta ERD in the same regions compared to younger adults. Furthermore, it has been shown that during simple movements, these beta-band oscillations reliably peak in the precentral gyri bilaterally with stronger activity contralateral to movement, while more complex movements (and some simple movements) also induce activity in the supplementary motor area and bilateral premotor cortices, postcentral gyri, parietal cortices, and cerebellum. Perhaps, it would be necessary to discuss about this in the intro and discussion section.

Answer: We agree with the reviewer’s comment. We have added a discussion about a significance of peri-and post-movement beta oscillations in the intro section. However, in the current study we were mostly focused on the pre-movement neuronal activity (after the audio cue and before mu- and beta-band ERD) and the provided statistical analysis in spatio-temporal domain did not reveal any significant age-related changes in cortical activation besides the increased theta-band activity in central-parietal sensors within this time frame. Maybe, this could be a consequence of experimental design and, particularly, a quite simple motor task, which execution may not strongly involve complex motor planning operations usually associated with early beta-band ERD.

Results

Comment: On the contrary, the between-subject differences were found in the spatial cluster, which included Cp3, Cpz, and Cp4 sensors (dorsal stream region of the sensorimotor area). Did you carry out a source analysis or else how are you actually talking about Ventral and Dorsal regions? I thought you have restricted your analysis at the sensor level. I wouldn’t talk about network based on sensor based functional connectivity in the time or frequency domain. Spatial resolution is too poor to argue for this even if you observed spectro-temporal signatures. For example, you mention distributed frontoparietal network. If I ask you how many areas do you think FPN comprise of then how would you answer? Please stick to sensor level analysis of your brain signals and connectivity patterns among sensor groups to discern changes. I don’t agree that talking about brain areas, ventral and dorsal streams etc. and above all networks makes much sense. You would probably admit networks in this case is loosely defined. This is also not a MEG Study.

Answer: We agree with the reviewer’s opinion. We have modified the Results section by sticking to the sensor-level description of the obtained results and excluding misleading formulations.

Discussion

Comment: I enjoyed reading the discussion section. Having said that I think the authors need to seriously look at some of their assertions. One thing is to see a change and other is to speculate about those change. Many of the formulations and interpretations are at this point remain unverified and speculative. Not a clear demonstration yet. However the good thing is that the authors have cited and covered references which are most relevant to their findings based on mainly two types of analysis time-frequency and functional connectivity. Further, they discussed various mechanisms which render support to their empirical observations and relates to existing literature. In particular, this is one of the key reason why instead of listing a series of plausible mechanisms, the authors could identify and evaluate quantitatively one or two key mechanisms highlighted in the discussion section. For example, the observation of increase in theta band activity/power in the Frontal sensors and in the sensors near sensorimotor areas could be sensorimotor integration (provided by Bland’s model) mechanisms or this could be long range connectivity change between distal brain areas leading to an elevation in the theta band power. Also, I don’t like the fact that you keep mentioning throughout the article about brain areas while you actually carry out all of your analysis primarily at the EEG sensor level. They clearly do not commensurate with each other. Please, see my previous comments on this issue persistent in this manuscript.

Answer: First, we would like to thank the reviewer for a positive feedback to our discussion section. Here, we also agree with the reviewer’s opinion that many interpretations and formulations are rather speculative. In the revised version of the Manuscript, we have tried to clarify and modify the most unsuccessful conclusions and formulation, to be less speculative and mostly associated with our observations.

Minor

Comment: Line number 190 may have a typo. Please check. Same pattern of typo recurring throughout the manuscript. Actually spotted them in several other places in this manuscript. Please revisit those sections where these typos are present and rectify.

Answer: We guess, that this typo is a ‘pre-motor phase’. We have corrected this typo by replacing it with a ‘pre-movement phase’ or ‘motor initiation phase’.

---

## [Editor Report · Decision Letter 1]

2 Sep 2020

Age-related slowing down in the motor initiation in elderly adults

PONE-D-20-14466R1

Dear Dr. Frolov,

We’re pleased to inform you that your manuscript has been judged scientifically suitable for publication and will be formally accepted for publication once it meets all outstanding technical requirements.

Kind regards,

Mukesh Dhamala, Ph. D.

Academic Editor

PLOS ONE

---

## [Editor Report · Acceptance letter]

4 Sep 2020

PONE-D-20-14466R1 

Age-related slowing down in the motor initiation in elderly adults 

Dear Dr. Frolov:

I'm pleased to inform you that your manuscript has been deemed suitable for publication in PLOS ONE. Congratulations! Your manuscript is now with our production department. 

Kind regards, 

on behalf of

Dr. Mukesh Dhamala 

Academic Editor

PLOS ONE